# Clinical and Molecular Diagnosis of Beckwith-Wiedemann Syndrome with Single- or Multi-Locus Imprinting Disturbance

**DOI:** 10.3390/ijms22073445

**Published:** 2021-03-26

**Authors:** Laura Fontana, Silvia Tabano, Silvia Maitz, Patrizia Colapietro, Emanuele Garzia, Alberto Giovanni Gerli, Silvia Maria Sirchia, Monica Miozzo

**Affiliations:** 1Department of Health Sciences, Medical Genetics, Università degli Studi di Milano, 20142 Milano, Italy; silvia.sirchia@unimi.it (S.M.S.); monica.miozzo@unimi.it (M.M.); 2Department of Pathophysiology and Transplantation, Medical Genetics, Università degli Studi di Milano, 20122 Milano, Italy; silvia.tabano@unimi.it (S.T.); patrizia.colapietro@unimi.it (P.C.); 3Laboratory of Medical Genetics, Fondazione IRCCS Ca’ Granda Ospedale Maggiore Policlinico, 20122 Milano, Italy; 4Clinical Pediatric Genetics Unit, Pediatrics Clinics, MBBM Foundation, S. Gerardo Hospital, 20900 Monza, Italy; smaitz@fondazionembbm.it; 5Istituto di Medicina Aerospaziale “A. Mosso”, Aeronautica Militare, 20138 Milano, Italy; emgarzi@tin.it; 6Reproductive Medicine Unit, ASST Santi Paolo e Carlo, Università degli Studi di Milano, 20142 Milano, Italy; 7Management Engineering Tourbillon Tech SRL, 35100 Padova, Italy; alberto@albertogerli.it; 8Research Laboratories Coordination Unit, Fondazione IRCCS Ca’ Granda Ospedale Maggiore Policlinico, 20122 Milano, Italy

**Keywords:** Beckwith-Wiedemann syndrome, multilocus imprinting disturbance, discordant monozygotic twins, X-chromosome inactivation, clinical diagnosis, molecular testing

## Abstract

Beckwith-Wiedemann syndrome (BWS) is a clinically and genetically heterogeneous overgrowth disease. BWS is caused by (epi)genetic defects at the 11p15 chromosomal region, which harbors two clusters of imprinted genes, *IGF2*/*H19* and *CDKN1C*/*KCNQ1OT1*, regulated by differential methylation of imprinting control regions, *H19/IGF2*:IG DMR and *KCNQ1OT1*:TSS DMR, respectively. A subset of BWS patients show multi-locus imprinting disturbances (MLID), with methylation defects extended to other imprinted genes in addition to the disease-specific locus. Specific (epi)genotype-phenotype correlations have been defined in order to help clinicians in the classification of patients and referring them to a timely diagnosis and a tailored follow-up. However, specific phenotypic correlations have not been identified among MLID patients, thus causing a debate on the usefulness of multi-locus testing in clinical diagnosis. Finally, the high incidence of BWS monozygotic twins with discordant phenotypes, the high frequency of BWS among babies conceived by assisted reproductive technologies, and the female prevalence among BWS-MLID cases provide new insights into the timing of imprint establishment during embryo development. In this review, we provide an overview on the clinical and molecular diagnosis of single- and multi-locus BWS in pre- and post-natal settings, and a comprehensive analysis of the literature in order to define possible (epi)genotype-phenotype correlations in MLID patients.

## 1. Introduction

Beckwith-Wiedemann syndrome (BWS; OMIM #130650) is an overgrowth disease characterized by macrosomia, macroglossia, lateralized overgrowth, visceromegaly, abdominal wall defects, and an increased risk of embryonal tumors. First described by Beckwith in 1963 and Wiedemann in 1964, BWS shows clinical and genetic heterogeneity. It is a pan-ethnic condition, with an estimated incidence of 1:10,000–13,700 live births [1,2] and an overall equal incidence in males and females, except for monozygotic twins, in which there is a female prevalence.

The incidence of the syndrome is probably underestimated given the existence of undiagnosed individuals with mild phenotypes, often related to the mosaic pattern of the underlying defect, that can, thus, be detected only in a subset of cells. BWS is caused by different epigenetic or genetic defects that alter the expression of known imprinted genes mapping at the 11p15 chromosomal region. This region harbors two imprinted domains, *IGF2/H19* and *CDKN1C/KCNQ1OT1*, regulated by the differential methylation in the two imprinting control regions (ICs), namely *H19/IGF2*:IG DMR (IC1) and *KCNQ1OT1*:TSS DMR (IC2) [3,4]. Among the alterations responsible for BWS, hypomethylation of IC2 is the most frequent. It can be present as single epimutation or, in a subset of cases, the methylation defect can involve additional imprinted loci, a condition called multi-locus imprinting disturbances (MLID) [5,6,7,8].

Moreover, the identification of a high incidence of female monozygotic (MZ) twins with discordant phenotypes and a high frequency of BWS among babies conceived by assisted reproductive technologies (ART), provide additional insights into the timing of imprint establishment during embryo development [9,10,11].

In this review, we provide an overview on the clinical and molecular diagnosis of single- and multi-locus BWS in a pre- and post-natal setting, by a comprehensive analysis of the literature data in order to further explore (epi)genotype-phenotype correlations in MLID patients compared with those with single-locus alterations.

## 2. Genetic and Epigenetic Alterations in BWS

### 2.1. (Epi)Genetic Defects at 11p15

The two imprinted domains at the 11p15 chromosomal region, spanning about 1 megabase, are regulated by two ICs, *H19/IGF2*:IG DMR (IC1) and *KCNQ1OT1*:TSS DMR (IC2), respectively (Figure 1).

IC1 regulates two imprinted genes: the paternally expressed *IGF2,* encoding a growth factor, and the maternally expressed *H19,* a lncRNA that functions as a tumor suppressor. In normal conditions, the IC1 is unmethylated and bound by the CTCF (CCCTC-binding factor), which acts as a chromatin insulator preventing *IGF2* promoter activation mediated by the enhancer downstream the *H19* gene. This setting results in silencing of *IGF2* and expression of *H19* (Figure 1). Differently, on the paternally derived allele, IC1 is methylated and cannot bind the CTCF, ensuing in the *IGF2* expression and *H19* silencing [12] (Figure 1).

IC2 controls several imprinted genes: *KCNQ1*, encoding a voltage-gated potassium channel, which is maternally expressed during early embryogenesis, but becomes biallelically expressed during fetal development [13]; *KCNQ1OT1*, which is a paternally expressed lncRNA antisense to *KCNQ1* [14]; and the maternally expressed *CDKN1C*, encoding a G1 cyclin-dependent kinase inhibitor that functions as a tumor suppressor [15]. IC2 maps at the *KCNQ1OT1* transcription start site and includes the promoter region of *KCNQ1* [16]. In normal conditions, IC2 on the maternal allele is methylated and leads to *KCNQ1OT1* silencing and *KCNQ1* and *CDKN1C* expression. On the paternally inherited allele, the *KCNQ1OT1* promoter is unmethylated and, consequently, the lncRNA is expressed and *KCNQ1* and *CDKN1C* are silenced [17] (Figure 1).

Most BWS patients (about 50–60%) show loss of methylation (LOM) at IC2, which results in a reduced expression of the *CDKN1C* gene on the maternal allele [19]. Gain of methylation (GOM) at IC1 on the maternal allele accounts for about 5–10% of BWS cases and leads to the downregulation of *H19* and overexpression of *IGF2* [19] (Figure 1).

The majority of the methylation defects in BWS patients are primary epimutation and occur in a mosaic pattern. However, up to 20% of patients with IC1 GOM may carry small copy number variants (CNVs) or single nucleotide variants (SNVs), resulting in IC1 GOM, while IC2 deletions and SNVs, leading to IC2 LOM, are very rare [20].

Paternal uniparental disomy of chromosome 11 (pUPD11) is the presence of two copies of the paternal 11p15 chromosomal region in absence of the maternal contribution. pUPD11 is observed in 20% of BWS patients [19], and results in the overexpression of *IGF2* and downregulation of *CDKN1C* (Figure 1). pUPD11 is mainly segmental and attributable to errors in postzygotic mitotic recombination, during early embryogenesis.

BWS can also be caused by mutations of the *CDKN1C* gene on the maternal allele, which are observed in about 5% of sporadic BWS and in 40% of familial cases [19]. In this latter occurrence, BWS phenotype follows an autosomal dominant segregation with incomplete penetrance, depending on the parental origin of the mutated allele. The risk of affected offspring is 50% when the mother is a carrier (Figure 1).

Finally, chromosomal abnormalities, including duplications, deletions, and translocations involving the 11p15 region, are detected in less than 5% of BWS cases [19]. In about 20% of BWS patients, the causative (epi)genetic alteration remain undetected [21].

### 2.2. Multi-Locus Imprinting Disturbance (MLID)

Although the 11p15 region is considered the disease-specific locus, a subset of BWS patients show aberrant methylation affecting also other imprinted loci, a condition described as multi-locus imprinting disturbance. MLID is not restricted to BWS, since it has been reported in other genomic imprinting diseases [22]. In BWS, MLID frequency is observed in 20–50% of patients with IC2 LOM [6,7,8,23,24,25,26,27,28,29,30,31,32], while it is very rare among cases with IC1 GOM [33]. Recently, genome-wide epigenetic defects, comprising both imprinted and non-imprinted loci, have been identified in patients with clinical diagnosis of BWS not confirmed by molecular testing. This finding expands the importance of MLID evaluation in the BWS diagnosis [34].

The etiology of MLID remains to be fully elucidated, as well as the mechanisms underlying the co-regulation of imprinting marks across the genome. MLID causative mutations in the genes encoding members of the NLRP protein family have been identified in a few BWS patients [35,36]. However, the role of these proteins in the regulation of the imprinting process remains to be clarified. *NLRP* mutations associated with a general hypomethylation cause hydatidiform mole [37]. Nevertheless, maternally-derived *NLRP* mutations identified in BWS patients with MLID suggests a different mechanism of pathogenicity compared to molar pregnancies [35,36], since in BWS, a heterogeneous pattern of aberrant methylation has been observed with a variable distribution of the involved loci.

*NLRP* variants in MLID cases may be associated with an uncommon mode of inheritance, the so-called “maternal effect,” in which gene variants can be present only in the unaffected mother, but they cause aberrant imprinting in the *NLRP* wild-type offspring [38]. NLRP proteins are components of the subcortical maternal complex (SCMC), playing a key role in imprinting establishment, before full zygotic genome activation [7]. In addition to *NLRP* genes, other loci belonging to the SCMC with a maternal effect have been investigated in families with MLID offspring, including *PADI6, OOEP, UHRF1* and *ZAR1* and an excess of variants in the *NLRP2*, *NLRP7* and *PADI6* genes have been reported in mothers of MLID children [7].

The identification of MLID provides new insights into the mechanisms of imprinting establishment and on the pathomechanisms underlying BWS development. The presence of MLID supports, indeed, the hypothesis of imprinting gene networks involved in the cross-talk among imprinted loci, probably mediated by complex chromatin structures and ncRNAs [39,40]. This hypothesis is also supported by recent evidence in BWS patients on the loss of the chromatin structure of the *IGF2/H19* and *CDKN1C/KCNQ1OT1* domains, which disrupts the fine-tuned control of gene expression in the two clusters [41]. The chromatin architecture of these domains differs between the maternal and paternal allele, and these allele-specific structures are required for the correct expression of the imprinted genes within these domains. In healthy cells, the *IGF2/H19* and *CDKN1C/KCNQ1OT1* domains fold in complex chromatin conformations that facilitate the control of imprinted loci mediated by distant enhancers. In BWS cells, the profound alterations of such chromatin architectures lead to the loss of the cross-talk between the two domains and the impairment of the correct expression of imprinted genes [41].

## 3. Clinical Features and Diagnosis

### 3.1. Prenatal Clinical Diagnosis

BWS clinical features may be evident at birth, although some clinical signs can also be identified prenatally. An early diagnosis of BWS is essential to plan patients’ management and interventions and to promptly start a tailored clinical follow-up and the monitoring for BWS-associated malignancies. A prenatal diagnosis of BWS can be reliably set in presence of two major features, including macroglossia, macrosomia, abdominal wall defects, or one major feature plus two minor findings, such as polyhydramnios, nephromegaly/kidney dysgenesis, adrenal mass suggestive of adrenal cytomegaly and placental mesenchymal dysplasia [42,43,44] (Table 1). These phenotypic signs can be ascertained by ultrasound imaging from the second trimester of gestation [45,46]. In addition, mothers of BWS fetuses have an increased risk (two–six-fold risk) of developing diabetes mellitus, gestational hypertension and proteinuria (suggestive of preeclampsia), compared to other pregnant women [47]. In BWS pregnancies, a high incidence of preterm delivery is also reported [48].

Recently, Shieh and colleagues [49] revised the ultrasound or MRI images in a cohort of prenatally diagnosed BWS cases in order to identify prenatal predictors of BWS suspicion and their evolution during pregnancy. This study evidenced that the ascertainment of a small omphalocele is the most frequent prenatal finding of BWS observed early in gestation (18–20 weeks), and confirmed postnatally with 100% of accuracy. Other BWS associated findings comprising macroglossia, macrosomia, visceromegaly and retrognathia can be detected later in gestation. Finally, high serum levels of alpha-fetoprotein have been associated with omphalocele [50].

According to the literature, serial ultrasound images are recommended for all fetuses showing an early small omphalocele, to evidence additional BWS signs during gestation. Furthermore, molecular prenatal diagnosis should be always offered to families with ascertained BWS history or in presence of signs associated with BWS [42].

### 3.2. Postnatal Clinical Presentation

Diagnosis of BWS is primarily based on the clinical evaluation and the syndrome was initially associated with the major findings of macrosomia, macroglossia and visceromegaly and abdominal wall defects. Subsequently, the phenotypic heterogeneity of BWS led to the definition of specific criteria for the clinical diagnosis. Accordingly, a BWS clinical diagnosis is made in the presence of three major clinical features or two major findings and one minor finding [3] (Table 2). Among clinical signs, macroglossia is the most common and it is observed in up to 97% of patients. Other frequent findings include abdominal wall defects, comprising omphalocele, umbilical hernia and diastasis recti, overall observed in about 80% of patients, while macrosomia is observed in 84%, hemihypertrophy in 64%, and outer ear anomalies (mainly posterior helical pits) are found in about 63% of cases. Minor signs comprise a number of pre- and post-natal clinical features with an overall frequency of about 50% (Table 2).

Despite the usefulness of the generally accepted criteria for BWS diagnosis, the phenotypic heterogeneity of the syndrome prompted the revision of the clinical scoring for BWS and the definition of the Beckwith-Wiedemann spectrum (BWSp) [51]. Accordingly, BWS patients are stratified in three subgroups: classic, with isolated lateralized overgrowth (ILO), and with atypical phenotypes. The BWSp scoring system classifies the clinical features in “cardinal” and “suggestive” (Table 3). Cardinal features are more likely to lead to a positive diagnosis and include macroglossia, exomphalos, lateralized overgrowth, multifocal Wilms tumors or nephroblastomatosis, hyperinsulinism and specific pathology findings (such as adrenal cytomegaly or placental mesenchymal dysplasia). Some of these features can be also prenatally detected. Suggestive features include a birthweight >2 standard deviations, facial nevus flammeus, polyhydramnios or placentomegaly, ear creases or pits, transient hypoglycemia, embryonal tumors, nephromegaly or hepatomegaly and umbilical hernia or diastasis recti. These signs can also be observed in the general pediatric population, but when present concomitantly with BWSp cardinal features, the clinical diagnosis of BWS is strongly sustained. Some features, included in the Weksberg diagnostic criteria (e.g., cleft palate, advanced bone age, polydactyly and supernumerary nipples), could be suggestive of alternative diagnoses and, therefore, have not been included in the BWSp scoring system.

Recently, a revision of the consensus has been proposed after the application of the BWSp scoring system to a large cohort of BWS patients [52]. In particular, the observation of an increased incidence of pUPD11 and IC1 GOM in the atypical and ILO groups suggests that the current criteria are mainly focused on the diagnosis of IC2 LOM patients and are less effective for the diagnosis of patients with other molecular defects. Furthermore, the frequent observation of unifocal/unilateral Wilms tumors or other embryonal neoplasms among BWS patients suggests moving these features into the cardinal group of clinical signs. Finally, patients with ILO and no other features should be included in the BWSp, and patients presenting with hyperinsulinism or a BWS-associated tumor should be evaluated for the subtle asymmetry or other BWSp suggestive features [52].

The BWSp scoring system also includes the recommendations and the impact of molecular testing in the diagnosis [51] (Figure 2). When the score is ≥4, the diagnosis is clearly defined and the molecular confirmation is not required. Cases with a score <2 do not meet the BWSp criteria, and therefore, the genetic testing is not indicated. Molecular testing is, instead, recommended for patients with score ≥2. When the molecular investigations are negative, a differential diagnosis should be considered. In patients with a score <4 and without molecular confirmation, a definitive clinical diagnosis of BWS cannot be set (Figure 2).

Finally, molecular testing remains mandatory for patients with a family history of BWS and a known heritable defect at the 11p15 chromosomal region [51].

### 3.3. Differential Diagnosis

A differential diagnosis should always be considered in presence of clinical features not commonly associated with BWS, since several overgrowth syndromes enter in differential diagnosis with BWS. In particular, the Simpson-Golabi-Behmel syndrome (SGBS, OMIM #312870) shows a clinical presentation partially overlapping with BWS (Table 4). SGBS patients present with macrosomia, visceromegaly, macroglossia and renal cysts. However, specific clinical signs, including the distinctive and coarse face, cleft lip, cardiac defects, supernumerary nipples, polydactyly and skeletal anomalies, may address the diagnosis of SGBS [54]. In addition, both SGBS and BWS patients have an increased risk for embryonal neoplasms, including Wilms tumors. SGBS is a recessive X-linked disease caused by mutations in the *GPC3* (glypican-3) gene. Some evidence also suggests a role of the *GPC4* gene, which maps close to *GPC3* [55,56].

Sotos syndrome type 1 (SOTOS1, OMIM #117550) is an overgrowth syndrome with some overlapping features with BWS. SOTOS1 patients are characterized by pre- and post-natal overgrowth, macrocephaly, variable intellectual disability and distinctive facial features, including prominent forehead with receding hairline, palpebral downslanting, pointed chin and advanced bone age [57] (Table 4). Dominant *NSD1* (Nuclear receptor-binding Set Domain protein 1) mutations account for about 60% of SOTOS1 cases; very rarely, patients with Sotos clinical features and pUPD11, or cases with *NSD1* mutations and clinical diagnosis of BWS with additional intellectual disability, have been reported [58].

Perlman syndrome (PRLMNS, OMIM #267000) is an autosomal recessive congenital overgrowth syndrome with similarities to BWS and characterized by neonatal overgrowth, hypotonia, organomegaly, characteristic facial dimorphisms (including inverted V-shaped upper lip, prominent forehead, deep-set eyes, broad and flat nasal bridge and low-set ears), renal anomalies (nephromegaly and hydronephrosis), neurodevelopmental delay and a high risk of neonatal mortality [59] (Table 4). Similar to BWS, PRLMNS is associated with a high risk of Wilms tumors and nephroblastomatosis, which is considered a precursor lesion of Wilms tumors. PRLMNS is caused by homozygous or compound heterozygous mutations in the *DIS3L2* gene. In addition, a neonatal case with clinical and molecular diagnosis of BWS, but with signs of both BWS and PRLMNS, was reported [60].

RASopathies, including Costello syndrome (CSTLO, OMIM #218040), may also enter in differential diagnosis with BWS. RASoptahies are caused by mutations in genes encoding components or regulators of the Ras/MAPK pathway, key regulators of the cell cycle, growth and differentiation. Each RASopathy exhibits a unique phenotype, but owing to the shared deregulated molecular pathway, they present with overlapping clinical features, including craniofacial dysmorphisms, cardiac defects, cutaneous, musculoskeletal and ocular abnormalities, neurocognitive impairment, hypotonia and an increased tumor risk [61]. RASopathies share some clinical features with BWS, such as macroglossia, macrosomia and increased tumor risk, but each RASopathy shows typical clinical features that distinguish it from BWS (Table 4).

*PIK3CA*-Related Overgrowth Syndromes (PROS) are a group of overgrowth disorders with a mosaic pattern and clinical features overlapping with BWS. PROS show, indeed, a variable phenotypic presentation characterized by overgrowth, lateralized overgrowth and vascular defect, including nevi [62] (Table 4).

Finally, it must be taken into account that lateralized overgrowth is a clinical feature shared by several syndromes: BWS, Kippel-Trenaunay syndrome, Proteus syndrome, McCune-Albright syndrome, epidermal nevus syndrome, triploid/diploid mixoploidy, Maffucci syndrome and osteochondromatosis [63].

### 3.4. Clinical Management and Follow-up

The clinical management of BWS patients requires a multidisciplinary approach, and an early diagnosis allows adequate follow-up and medical intervention in order to avoid related complications. The identification of macrosomia in the prenatal period, for example, increases the risk of trauma during delivery, such as cephalohematoma, brachial plexus injury and respiratory distress syndrome, thus prompting the choice of a cesarean section [64].

Macroglossia, which is the most frequent clinical sign observed in BWS, can lead to feeding and breathing complications in infancy, and it can affect speech and lead to malocclusion, with the consequences of misalignment of the teeth and prognathism, in addition to esthetic concerns. For these reasons, BWS patients with macroglossia should be monitored by a multidisciplinary medical team, including plastic/maxillofacial surgeons, pulmonologists, orthodontists and speech specialists, in order to consider, case by case, the effectiveness of tongue surgical resection and jaw reduction, and the impact of this congenital defect on breathing, language and swallowing [65,66].

In presence of hypoglycemia or hyperinsulinemia, the early intervention and the referring to an endocrinologist is mandatory. In addition, patients with lateralized overgrowth should be evaluated by an orthopedist, and cardiac and renal evaluation is also recommended to identify possible congenital defects [51].

A major concern regarding BWS follow-up is tumor screening. BWS patients have a higher risk of developing embryonal tumors, in particular Wilms tumors and hepatoblastoma. The estimated overall risk for malignancies in BWS is 7.5%, with a higher risk at birth that reaches the general population risk before puberty [67]. BWS patients should undergo routine tumor surveillance with abdominal/renal ultrasounds and the serial dosage of serum alpha-fetoprotein (AFP), even if its utility is still debated. However, indications for tumor surveillance differ among centers, with some performing screening of all BWS patients, while others recommend tumor surveillance only for patients with a higher risk according to (epi)genotype-phenotype correlations (see Section 5).

The European BWS consensus [51] recommends tumor screening for all BWS patients (also without molecular diagnosis) with the exclusion of patients with IC2 LOM, advising only abdominal ultrasound every three months from the time of diagnosis until seven years of age [51]. More recently, Duffy et al. [52] provided insights into the practical diagnosis and management recommendations on the basis of the analysis of a large cohort of patients with BWSp. They suggested tumor screening for all BWS patients with both abdominal ultrasound and AFP dosage every three months until four years of age and renal ultrasound every three months until seven years of age [52].

For monozygotic (MZ) twins discordant for BWS, since the unaffected twin may show a broad range of phenotypic presentation [9,68], the clinical follow-up of the healthy twin has been long debated, until recently a specific algorithm for the clinical management of multiple pregnancies (in which at least one child is affected by BWS) was proposed [69]. According to this algorithm, in monozygotic gestations (monochorionic or dichorionic), both twins should be clinically evaluated according to the BWSp scoring system, but molecular testing should be performed only if the twin show a score ≥4 and at least one cardinal feature. Tumor screening is recommended for the affected twin but not for the discordant twin, if this latter does not present any clinical features, unless positive molecular testing in tissues other than blood [69].

## 4. Molecular Diagnosis

### 4.1. Postnatal Testing

Molecular testing for BWS is recommended for patients with a score ≥2 and methylation analysis of IC1 and IC2 should be performed first (Figure 2). Aberrant methylation levels at a single IC are indicative of a primary epimutation or, rarely, of CNVs, while the identification of both IC1 GOM and IC2 LOM may be associated with segmental pUPD11. Methylation profiling of the 11p15 region may be performed using different techniques, including bisulfite pyrosequencing and methylation-specific multiplex ligation-dependent probe amplification (MS-MLPA) [70,71], which allows the simultaneous detection of IC1 and IC2 methylation pattern and the copy number status. A positive methylation test, with techniques that do not implement CNV analysis, should be followed by CNV testing to rule out the presence of deletions/duplications/translocations, resulting in aberrant methylation of the ICs (Figure 2). In presence of a CNV, this should be characterized by chromosome microarray, FISH or karyotyping [53,72]. Furthermore, in presence of IC1 GOM, IC1 sequencing should be performed to detect the possible presence of SNPs or small deletions, resulting in aberrant methylation levels [73,74]. SNParray or microsatellite analysis is also mandatory to confirm and characterize pUPD11 (Figure 2).

Mutational analysis of *CDKN1C* by direct sequencing is performed in patients with a normal methylation profile at IC1 and IC2 (Figure 2). In the presence of CNVs or mutations, the analysis should be extended to other family members to define the inheritance pattern and calculate the risk of recurrence.

It must be taken into account that a negative result of methylation tests does not exclude the diagnosis of BWS, since a low-level of mosaicism, with variable presentation in different tissues, can be present. Therefore, testing of samples from other tissues besides blood, such as buccal swab or skin fibroblasts, may improve the diagnostic yield [5]. However, a further negative methylation test in other tissues is not conclusive; in these patients, a differential diagnosis should always be considered and appropriate tests performed [53].

Up to 50% of BWS patients with methylation defects, mainly IC2 LOM, show MLID and in a subset of them, the condition may be caused by mutations in genes involved in methylation establishment/maintenance. However, the clinical utility of MLID testing and the identification of associated mutations is still uncertain [5], mostly because specifically MLID associated clinical findings have not been yet identified. For this reason, MLID evaluation is currently performed only for research purposes (Figure 2).

### 4.2. Prenatal Testing

Methylation pattern at IC1 and IC2 is stable in the placenta [75,76]. The diagnosis of BWS can, thus, also be performed in prenatal cases using the same technical approaches described for postnatal molecular diagnosis. Prenatal diagnosis of BWS may be carried out in early pregnancy by the analysis of chorionic villus sampling (CVS), or later using both native or cultured amniocytes [44,77]. However, mosaicism complicates prenatal molecular diagnosis using CVS, as the mosaicism may be confined to the placenta or can involve the fetus or it can affect both. As a consequence, the possible heterogeneous distribution of the mutated cells in the placenta may lead to false-negative results. Another issue concerning prenatal diagnosis by CVS may be the incomplete set of methylation marks at the time of diagnosis [77]. However, methylation levels at IC1 and IC2 have been demonstrated to be already set in the placenta at early stages, thus confirming the reliability of the analysis of these loci for prenatal methylation testing of BWS by CVS. On the contrary, *H19* is hypomethylated in CVS, and this must be taken into account in performing prenatal diagnosis analyzing this locus [44].

Regarding the mutational analysis of *CDKN1C* in prenatal diagnosis, it should be performed only in cases of known pathogenic variants in the mother, because of the high frequency of polymorphic variants in the gene, often with unknown pathogenicity.

A negative result in prenatal molecular testing does not exclude the diagnosis of BWS and a postnatal analysis should always be performed.

## 5. (Epi)Genotype-Phenotype Correlations

### 5.1. Prenatal and Neonatal Correlations

Few studies investigated possible (epi)genotype phenotype correlations according to prenatal clinical features and/or placental defects. Despite that most of the BWS features cannot be recognized in utero, a clear association was observed between IC2 LOM or *CDKN1C* mutations and omphalocele [18,78,79]. In addition, also pUPD11 has been reported to be associated with omphalocele, with a direct correlation between the proportion of isodysomic cells and organomegaly [80].

Fetal and neonatal growth patterns may allow the subgroup of BWS cases according to the (epi)genotype. In detail, BWS fetuses with IC1 GOM show extreme macrosomia and a severe disproportion between weight and length, while in IC2 LOM and *CDKN1C* mutated neonates, macrosomia is less pronounced and the weight/length ratio is more proportionate [18]. pUPD11 babies show growth parameters close to the IC2 LOM group, but with body mass proportion similar to IC1 GOM cases, probably as a consequence of somatic mosaicism. Moreover, the gestational age of the delivery correlates with the molecular defects, and most of the preterm BWS babies carry *CDKN1C* mutation or IC2 LOM [18].

BWS is frequently associated with placental pathological findings, in particular placentomegaly (≈70% of cases), placental mesenchymal dysplasia (≈22% of cases), chorangioma/chorangiomatosis (≈23% of cases) and extravillous trophoblastic cytomegaly (≈22% of cases) [81]. However, no (epi)genotype-phenotype correlations related to placental defects in BWS have been clearly defined [81].

### 5.2. Postnatal Correlations in Single- and Multi-Locus Patients

Given the heterogeneous clinical presentation of BWS patients, several studies tried to determine correlations among clinical features and specific molecular defects at 11p15, in order to help clinicians in the classification of patients and in a tailored follow-up.

Studies on (epi)genotype-phenotype correlations in BWS are always ongoing and, to date, a strong association was defined between Wilms tumors and IC1 GOM or pUPD11. These latter patients also have a high incidence of hepatoblastoma [18,66,82]. Instead, omphalocele is associated with IC2 LOM and *CDKN1C* mutations [51] (Table 5).

Milder (epi)genotype-phenotype associations have been drawn for macroglossia, correlated with IC2 LOM or IC1 GOM [18,83], macrosoma, organomegaly and nephrourological anomalies that are preferentially observed in IC1 GOM patients [18,83,84,85], and lateralized overgrowth associated with pUPD11 [18,83,86] (Table 5). Finally, hypoglycemia is constantly present in IC1 GOM or pUPD11 patients [83,84] (Table 5).

Recently, an increased incidence of undescended testes in patients with IC1 GOM and IC2 LOM was observed [52], probably associated with the high frequency of prematurity in these groups. IC1 GOM patients show bilateral/multifocal Wilms tumors more frequently than patients with IC2 LOM or pUPD11 [52] (Table 5). Moreover, patients with IC2 CNVs, genomic rearrangements or *KCNQ1* variants may be predisposed to long QT syndrome [87,88,89].

No clear (epi)genotype-phenotype correlations nor specific clinical features have been defined so far in BWS patients with MLID [5]. MLID patients are not likely to present additional clinical signs not related to BWS. However, a high incidence of females and a high frequency of some clinical features was reported in MLID patients [8]. Notably, most BWS MLID cases show IC2 hypomethylation.

Herein, we performed a comprehensive analysis of literature data [5,8,23,24,28,33] of the clinical features of BWS patients with MLID (73 cases) compared to patients with single-locus defects (178 cases with IC2 LOM, IC1 GOM or pUPD11). Among single-locus defects, MLID was mainly reported in patients with IC2 LOM and only Maeda et al. found MLID in three patients with IC1 GOM [33]. We observed that MLID patients show a frequency of clinical features about 15% higher compared to single-locus ones (Figure 3), thus indicating that MLID cases show a more severe phenotype. In addition, the analysis between the two groups suggests that nevus flammeus, abnormal glycemic control and polyhydramnios are more frequent in BWS-MLID patients (difference between group ≥10%) (Figure 3). Only tumor incidence is higher in single-locus patients, probably as a consequence of the inclusion of patients with IC1 GOM in the analyzed studies. In addition, cardiac anomalies are the only clinical signs restricted to single-locus patients (Figure 3).

Finally, considering the sex distribution, our analysis shows a significant female prevalence in the MLID group (*p*-value = 0.0314 at Fishers’ exact test) (Figure 3), a finding strongly confirmed in MZ twins discordant for BWS. Although the reason for this association is still obscure, a possible link with X-chromosome inactivation (XCI) establishment cannot be excluded (see Section 6.1).

Taken together, these results suggest the need to expand the analysis to additional cases to delineate specific (epi)genotype-phenotype correlations in MLID cases and the importance of the implementation of molecular investigation to detect multi-locus alterations.

## 6. Deciphering the Timing of Epigenetic Errors Establishment

### 6.1. Discordant Monozygotic Twins and X-Chromosome Inactivation

The incidence of female monozygotic (MZ) twins among BWS patients is increased compared to the general population (2.5% vs. 0.3–0.4%) [10,90,91]. It is conceivable that this incidence is underestimated, since it is known that an initial twin pregnancy can end with the birth of a child, after the loss of a twin. In addition, and early fetal loss is 10 times more frequent in monochorionic than dichorionic twin pregnancies, possibly due to the existence of pregnancy complications unique to monochorionicity, such as the twin-to-twin transfusion syndrome and a selective fetal growth restriction [92]. From this evidence, we hypothesize that a subset of BWS females might derive from initial twin pregnancies.

BWS MZ twins are frequently females and phenotypically discordant, despite sharing the 11p15 alteration. IC2 LOM is the most frequent defect observed in discordant twins, and it is usually shared in the blood of both twins, probably as a consequence of placental vascular connections [10,90,93,94]. In addition, MLID is common in discordant MZ twins [90,94], with approximately 50% of MZ twins with IC2 LOM and methylation defects extended at other imprinted loci [90].

Overall, these results are in line with the higher frequency of females among MLID patients (see Section 5.2) and support a link between gender, twinning and BWS/MLID. Several studies have tried to address this issue and investigated the timing of the imprinting error establishment. Bliek and colleagues [90] proposed a mechanism in which the methylation failure precedes, and possibly triggers, the twinning process itself. Further studies led to the definition of the theory of the “diffuse mosaicism” [69], according to which the epigenetic event leading to BWS triggers the twinning process and affected cells can diffuse in the twin embryos, leading to a mosaic distribution of the cells carrying the epimutation. The variable degree of phenotypic discordance in BWS multiple pregnancies has, thus, been related to the degree of the mosaicism, which, in turn, is associated with the timing of the twinning event and the establishment of the epigenetic defect [69].

The increased prevalence of female twins in the BWS population suggests that the mechanism that triggers the epigenetic processes related to post-zygotic imprinting setting could also force the twinning process, specifically in the presence of an XX chromosome complement. In particular, it has been proposed that the lag in early development of female embryos, secondary to XCI, makes female MZ twin embryos more prone to imprinting errors [63,95]. In addition, since imprinting establishment has features in common with XCI, defects in this latter process may also affect autosomal imprinting. As a consequence, since XCI occurs only in the presence of more than one X chromosome, some imprinting defects may occur mainly in females [95]. Evidence that supports this hypothesis comes from animal models and the Bestor theory [96], suggesting that failure of the DNMT1o (the oocyte form of DNMT1) to maintain IC2 methylation may lead to epigenetic asymmetry and the separation, by the twinning event, of cell populations with different IC2 methylation patterns. DNMT1o is also involved in the XCI process, possibly linking nonrandom X-inactivation to BWS discordance [97]. Although many cases of MZ twins discordant for BWS were published [10,90,93,94], no definitive conclusions have been drawn on the mechanisms linking female, twinning and discordance for BWS/MLID.

We recently investigated a MZ female twin pair discordant for BWS by analyzing MLID and the XCI pattern. We found that the affected twin carried IC2 LOM and MLID in both blood and buccal swab, while the healthy twin presented IC2 LOM and MLID only in blood, probably as a consequence of placental vascular connections [11]. These results suggest a common pathomechanism leading to IC2 LOM and MLID that acts either after the twinning event and/or forces the segregation of the epimutated cells to one embryo. Regarding the XCI pattern, since both twins presented the same skewed XCI, we hypothesized that the establishment of XCI precedes and may trigger IC2 LOM and MLID [11]. Interestingly, deviations from the random pattern of XCI has been already reported in non-X-linked diseases [98].

To explore the link between XCI pattern and MLID, we assessed the XCI ratio using the Humara test, as previously reported [99], in MLID and single-locus (IC2 LOM) female of our population (*n* = 9), and found a slightly higher frequency of skewed XCI (≥75%) in MLID (60%) vs single-locus BWS females (50%). These values seem to be higher compared to the frequency of preferential XCI observed in the general population (about 30%) [99]. Although the importance to investigate other cases is clear, our preliminary findings support a possible link between a preferential XCI pattern and the epigenetic errors causative of IC2 LOM and MLID (see Section 5.2).

### 6.2. Assisted Reproductive Technologies (ART)

A four- to six-fold increased risk of BWS in ART pregnancies has been reported and most of these patients carry IC2 LOM [64,100,101,102]. This finding supports the proposal to include ART in the criteria for the diagnostic scoring of BWSp [52], and point to a critical role of the pre-implantation phase of the embryo for imprinting establishment/maintenance. To date, no specific ART procedures have been associated with an increased risk of BWS and all phases of ART, comprising ovarian stimulation, might influence genomic imprinting. In addition, it is not yet clear whether the high incidence of BWS in ART pregnancies is due to the related procedures or to the underlying reasons of infertility.

Further studies on BWS patients conceived by ART could clarify the impact of ART on imprinting disturbances and on the timing of the imprinting error occurrence.

## 7. Conclusions

BWS, although rare, is a disease extensively studied for its peculiar and complex clinical and molecular features. Recently, new relevant knowledge has emerged regarding the multi-locus pattern of the epigenetic alterations, which opens up new avenues of investigation at genomic level.

Although no specific clinical signs are probably present in MLID, we observed that MLID patients show a higher frequency of BWS-related clinical features compared to single-locus ones, thus suggesting that MLID patients frequently show a more severe phenotype. Rare maternal *NRLP* variants explain only a subset of MLID cases, but not, for example, BWS-MLID cases associated with ART.

Finally, the female prevalence among BWS-MLID cases in both singleton and twin pregnancies leads to the investigation on possible epigenetic mechanisms shared by genomic imprinting and XCI.

## Figures and Tables

**Figure 1 ijms-22-03445-f001:**
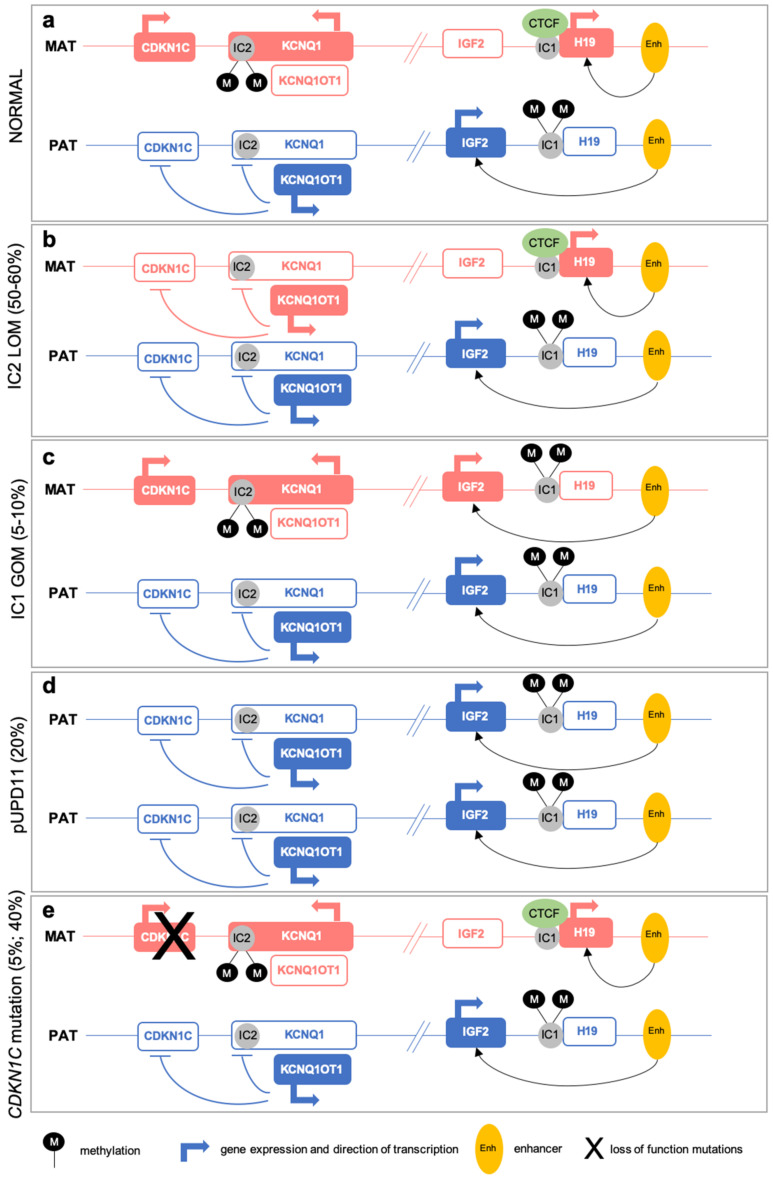
Schematic representation of genetic and epigenetic defects underlying Beckwith-Wiedemann syndrome (BWS). (**a**) Imprinting setting of IC1 and IC2 and expression of imprinted genes in the two 11p15 clusters on the maternal (red) and paternal (blue) allele in normal conditions. (**b**) IC2 LOM on the maternal allele leading to reduced expression of *CDKN1C* and *KCNQ1*. (**c**) IC1 GOM at the maternal allele leading to biallelic expression of *IGF2* and silencing of *H19*. (**d**) pUPD11 resulting in the downregulated expression of *CDKN1C* and *H19* and biallelic expression of *IGF2*. (**e**) Maternal *CDKN1C* loss of function mutations resulting in absence of functional CDKN1C protein (adapted with permission from [18], Copyright (2021) John Wiley and Sons).

**Figure 2 ijms-22-03445-f002:**
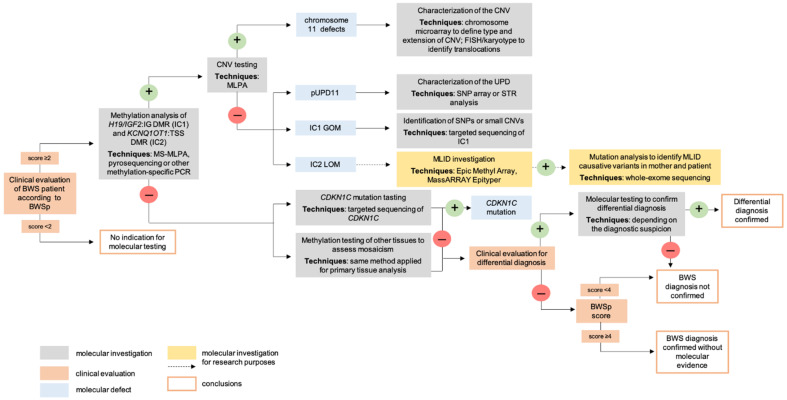
Diagnostic flowchart for BWS according to the BWSp scoring system [51]. Only patients with a diagnostic score ≥2 should receive molecular testing. Molecular testing, reported in grey boxes, should always start with a methylation analysis of IC1 and IC2, and in case of a positive result, it should always be followed by a copy number variant (CNV) assessment. A positive result (light blue boxes) of methylation analysis may derive from a primary epimutation at ICs (IC2 LOM or IC1 GOM) or pUPD11 (both IC2 LOM and IC1 GOM). SNP array may confirm pUPD11. Patients negative at methylation evaluation and CNV testing should undergo *CDKN1C* mutational analysis. In patients with IC2 LOM MLID methylation analysis and whole-exome sequencing for the identification of MLID causative mutations can be performed for research purposes. Negative results at all standard BWS testing may be due to tissue mosaicism and, in these cases, the analysis of tissue samples other than blood is mandatory. Differential diagnosis should be considered and appropriate tests performed. Patients with negative results for all the analyses, but with a clinical score ≥4, have a clinical diagnosis of BWS without a molecular evidence (adapted from [53]).

**Figure 3 ijms-22-03445-f003:**
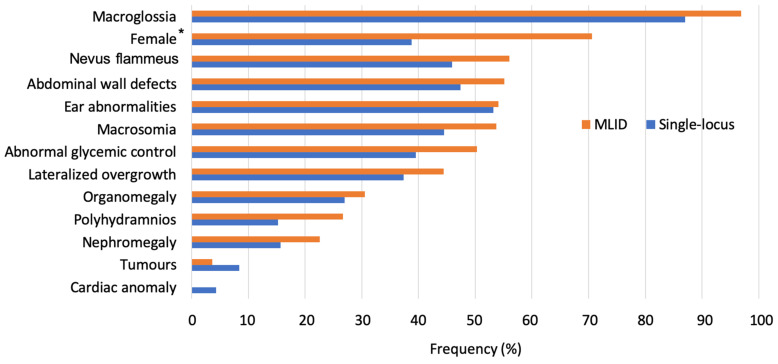
Frequency distribution of clinical features in BWS patients with MLID (orange) and with single-locus defects (blue) [5,8,23,24,28,33]. * *p*-value = 0.0314 at Fisher’s exact test.

**Table 1 ijms-22-03445-t001:** Prenatal clinical features associated with BWS.

Major Features	Minor Features
Macroglossia	Polyhydramnios
Macrosomia	Nephromegaly/kidney dysgenesis
Abdominal wall defect	Suprarenal mass suggestive for adrenal cytomegaly
	Placental mesenchymal dysplasia

**Table 2 ijms-22-03445-t002:** Diagnostic criteria for BWS according to Weksberg [3].

Clinical Feature	Frequency (%)
Major criteria	
Macroglossia	97
Macrosomia	84
Abdominal wall defects	80
Hemihypertrophy	64
Outer ear anomalies	63
Kidney and ureter anomalies	28–61
Visceromegaly	41
Embryonal tumors	~8
Cleft palate	~6
Positive family history	-
**Minor criteria**	
Nevus flammeus of the forehead	54
Neonatal hypoglycemia	>50
Prematurity	50
Placentomegaly	50
Polyhydramnios	50
Diastasis recti	28
Cardiomegaly/hypertrophic cardiomyopathy	20
Typical facies	-
Polydactyly	-
Supernumerary nipples	-
Advanced bone age	-

**Table 3 ijms-22-03445-t003:** Classification of BWS clinical signs according to the Beckwith-Wiedemann spectrum (BWSp) scoring system [51].

Cardinal Features (2 Points for Feature)
Macroglossia
Exomphalos
Lateralized overgrowth
Multifocal and/or bilateral Wilms tumors or nephroblastomatosis
Hyperinsulinism (lasting >1 week and requiring medical treatment)
Pathology findings: adrenal cortex cytomegaly, placental mesenchymal dysplasia or pancreatic adenomatosis
**Suggestive Features (1 Point for Feature)**
Birthweight >2SDS above the mean
Facial nevus flammeus
Polyhydramnios and/or placentomegaly
Ear creases and/or pits
Hyperinsulinism (lasting <1 week)
Typical tumors: neuroblastoma, rhabdomyosarcoma, unilateral Wilms tumors, hepatoblastoma, adrenocortical carcinoma or pheochromocytoma
Nephromegaly and/or hepatomegaly Umbilical hernia and/or diastasis recti

**Table 4 ijms-22-03445-t004:** Clinical features overlapping with BWS and syndromes that enter in differential diagnosis.

	BWS	SGB	Sotos	Perlman	RASopathies	PROS
Macroglossia	++	++			+	
Abdominal wall defects	++	++	+			
Segmental/lateralized overgrowth	++		+			++
Increased embryonal tumor risk	++	++		+ ^§^	+ ^^^	+
Neonatal hypoglycemia	++	++	+			
Birthweight > 2SD/macrosomia	+	++	++ ^*^	++	++	+
Vascular defects	+				+	++
Ear creases and/or pits	+	+			+	
Macrocephaly		++	++	++	++	
Developmental delay/intellectual disability		++	++	++	++	+ ^°^
Peculiar dysmorphic features	+	++	++		++	+/−
Congenital heart disease		++	++	+	++	+
Nephromegaly and/or hepatomegaly	+	++		+	+	+

^+^, present; ^++^, frequently present; ^+/−^, may be present or not; ^*^ birth weight usually normal; ^§^ Wilms tumor; ^^^ especially in Costello syndrome; ^°^ especially in megalencephaly.

**Table 5 ijms-22-03445-t005:** Molecular alterations observed in BWS patients, recurrence risk and (epi)genotype-phenotype correlations.

Molecular Alteration	Frequency	Mosaicism	Risk of Recurrence	(Epi)Genotype-Phenotype Correlations
IC1 GOM	5–10%	Yes	<1% without genetic anomalies; 50% depending on the parental origin and if genetic anomalies are present (up to 20% of SNVs)	Bilateral/multifocal Wilms tumors, macroglossia, macrosoma, organomegaly, nephrourological, hypoglycemia, undescended testes
IC2 LOM	50–60%	Yes	<1% without genetic anomalies; 50% depending on the parental origin if genetic anomalies are present	Omphalocele, macroglossia, undescended testes
pUPD11	20%	Yes	<1%	Lateralized overgrowth, Wilms tumors, hypoglycemia
*CDKN1C* mutations	5% sporadic cases; 40% familial cases	Rarely	50% via maternal transmission	Omphalocele
MLID	50% IC2 LOM patients; rare in patients with no molecular diagnosis	Yes	Low without causative mutations	Not clearly defined with an excess of female and a high frequency of some features

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
