# Peer review of "Clinical and Molecular Diagnosis of Beckwith-Wiedemann Syndrome with Single- or Multi-Locus Imprinting Disturbance"

_ijms, 2021, doi:10.3390/ijms22073445_

Round 1
Reviewer 1 Report
This is a well written, in broad lines accurate, exhaustive overview on the clinical and molecular diagnostics of primarily Bechwith Wiedeman syndrome. This review can, with some adaptations (see below) be used for educational purposes for students, clinical geneticists and pediatricians.
Specific comments
Line 117 The authors claim that MLID is observed in up to 50% of patients with IC2 LOM. This seems to be an overestimation as compared to frequency reported in other papers (~ 35%, see ref 7, Begeman et al. 2018).
Lines 126-138 The authors mention that variants in NLRP genes in mothers of children with ML imprinting disorders might be implicated in the pathogenesis of MLID. However they fail to mention additional genes that have been associated with this mechanism (see see ref 7, Begeman et al. 2018)
Lines 334 and 335: The authors state that tumor screening protocol suggested by the European BWS consensus is revised by ref 49 Duffy, K.A. et al. This group was one of the partners in the consensus meeting, and the protocol described reflects a different American view on screening itself. The protocol of the European consensus group is as far as I know unchanged.
Lines 561-562: The authors compare the clinical features of patients with MLID and single locus defects. It is not clear whether the latter include patients with only true single locus defects like GOM IC1 and with LOM of IC2 or whether pUPD11 patients (in fact a two locus defect) are also included. This term is also used in the title.
We would suggest comparing MLID patients with LOM IC2 patients, because they are known to fall in that diagnostic subgroup.
Defining the ‘single locus defects’ may also clarify the statement in 568-569: Interestingly, only tumor incidence is higher in single-locus patients, confirming the correlation between tumor development and IC1 GOM.
Lines 631-634: The authors state mention that that IC2 LOM and MLID showed a different tissue distribution between the twins. The difference in methylation in blood and buccal swab is not a real difference between tissues in one individual, but merely a result from vascular connections by which aberrant cells were transferred from the affected to the non-affected twin as is mentioned in line 598-599.
Reviewer 2 Report
This review manuscript contained large sections of summary on clinical and molecular studies on BWS, and towards the end, the authors added minor portion of reanalysis of literature data. The authors have published multiple original research article in this topic. The contents seem to be informative for the field as it provide comprehensive summary and authors’ proposal on mechanistic explanations of multi-locus imprinting and BWS. There were several points in which reference may be added.
- In line 85 and 87, there are specific numbers (about 50-60%, about 5-10%) without specific reference. It would be helpful to indicate where these numbers came from.
- In line 99, this paragraph also does not have reference.
- In line 112, is this part of reference 19?
- Line 125 was disrupted in the middle.
- In line 356, bisulphite > bisulfite?
- In Figure 3 and Table 6 show essentially the same information. It may be easier to read to show these two panels together or keep Table 6 only.
